# Increased Natural Killer Cells Are Associated with Alcohol Liver Fibrosis and with T Cell and Cytotoxic Subpopulations Change

**DOI:** 10.3390/jcm11020305

**Published:** 2022-01-08

**Authors:** Paola Zuluaga, Aina Teniente-Serra, Daniel Fuster, Bibiana Quirant-Sánchez, Anna Hernandez-Rubio, Eva Martínez-Cáceres, Roberto Muga

**Affiliations:** 1Department of Internal Medicine, Hospital Universitari Germans Trias i Pujol-IGTP, Universitat Autònoma de Barcelona, 08916 Barcelona, Spain; dfuster.germanstrias@gencat.cat (D.F.); ahernandezr.germanstrias@gencat.cat (A.H.-R.); rmuga.germanstrias@gencat.cat (R.M.); 2Department of Immunology, Hospital Universitari Germans Trias i Pujol-IGTP, Universitat Autònoma de Barcelona, 08916 Barcelona, Spain; ateniente.germanstrias@gencat.cat (A.T.-S.); bquirant.germanstrias@gencat.cat (B.Q.-S.); evmcaceres@gmail.com (E.M.-C.)

**Keywords:** alcohol liver disease, FIB-4 score, lymphocyte population

## Abstract

Natural killer (NK) cells play a therapeutic role in liver fibrosis (LF). We aimed to analyze NK cells in heavy drinkers without cirrhosis or decompensated liver disease and establish correlations with other related subpopulations. Data on sociodemographic characteristics, alcohol consumption, laboratory parameters, and immunophenotyping of NK (CD16^+^/CD56^+^), T (CD3^+^), B (CD19^+^), NKT (CD16^+^/CD56^+^/CD3^+^), and cytotoxic (CD3^-^CD8^+^) cells were collected. Fibrosis-4 (FIB-4) scores were used to compare patients without (FIB-4 < 1.45) and with (FIB-4 > 3.25) advanced LF (ALF). We included 136 patients (76% male) with a mean age of 49 years who had a 15-year alcohol use disorder (AUD) and alcohol consumption of 164 g/day. Patients with ALF (*n* = 25) presented significantly lower absolute total lymphocyte, T cell, B cell, and NKT cell numbers than patients without LF (*n* = 50; *p* < 0.01). However, the NK cells count was similar (208 ± 109 cells/µL vs. 170 ± 105 cells/µL) in both groups. The T cells percentage was lower (80.3 ± 5.6% vs. 77 ± 7%; *p =* 0.03) and the NK cells percentage was higher (9.7 ± 5% vs. 13 ± 6%; *p =* 0.02) in patients with ALF than in those without LF. The percentages of NK cells and T cells were inversely correlated in patients without (r = –0.65, *p* < 0.01) and with ALF (r = −0.64; *p* < 0.01). Additionally, the NK cells and CD3^-^CD8^+^ cell percentages were positively correlated in patients without (r = 0.87, *p* < 0.01) and with (r = 0.92; *p* < 0.01) ALF. Conclusions: Heavy drinkers without decompensated liver disease showed an increase in NK cells related to T cells lymphopenia and an increase in cytotoxic populations. The interaction of NK cells with other subpopulations may modify alcohol-related liver disease progression.

## 1. Introduction

Liver fibrosis (LF) is the main prognostic marker for the progression of alcohol liver disease (ALD) [1]. The mechanisms of hepatic inflammation as a triggering factor in stellate cell activation and in the consequent development of LF have been extensively studied [2]. Immune cells are key to the progression of ALD [3]. In particular, natural killer (NK) cells have been extensively studied for their potential antifibrotic effects and as therapeutic targets [4].

In addition to inhibiting hepatic stellate cells and preventing the progression of LF, NK cells have antiviral and antitumor effects through the secretion of regulatory cytokines and the release of cytotoxic granules [5]. In humans, NK cells represent 5–20% of peripheral lymphocytes and up to 50% of resident lymphocytes in the liver; however, their phenotype and function vary widely depending on their location [6].

Alcohol affects cytolytic activity and reduces the peripheral NK cells count [7]. Studies have demonstrated that chronic alcohol exposure alters the balance between NK cells derived from the bone marrow and the thymus [8]. Consequently, alcohol exposure affects development and maturation and decreases the availability of interleukin (IL)-15, a key cytokine in the NK cell cycle [9]. Moreover, the lack of activation of NK cells and the resistance of hepatic stellate cells to apoptosis are mechanisms related to the progression of ALD [10,11]. Despite this evidence, in vitro studies do not appear to adequately reproduce NK cell activation or the stellate cell microenvironment [12].

In healthy individuals exposed to alcohol, the cytotoxicity of NK cells is not altered [13]. Conversely, although patients with alcohol-related cirrhosis and severe acute alcoholic hepatitis present an increased peripheral NK cells count, there is less activation and capacity for degranulation [14,15]. A recent study found that patients with alcohol-related cirrhosis had a higher percentage of NK cells than did healthy individuals and patients with acute alcoholic hepatitis. However, the NK cells of the patients with alcohol-related cirrhosis showed greater cytotoxicity than those of healthy individuals, as measured by the release of cytotoxic granules [16]. The discrepancies between these results suggest that alcohol may exert different effects on NK cells based on the different stages of liver disease.

The distribution of NK cells in the early stages of ALD has not been previously described. Their correlation with other lymphocyte subpopulations may contribute to a better understanding of the pathogenesis of ALD. The main objective of this study was to analyze changes in the proportion of peripheral NK cells in heavy drinkers without evidence of decompensated liver disease or overt alcohol-related cirrhosis. The secondary objective was to analyze the relationships between NK cells and other lymphocyte subpopulations in the presence of LF.

## 2. Materials and Methods

### 2.1. Patients

This was a cross-sectional study that included 136 patients consecutively admitted for alcohol use disorder (AUD) treatment in a tertiary care setting between March 2013 and December 2018. The patients were diagnosed with AUD according to the Diagnostic and Statistical Manual of Mental Disorders (DSM-5) criteria [17]. The main criteria for admission were severity of the AUD, risk of severe withdrawal syndrome, and difficulty in following outpatient treatment. On the first day of admission, the patient’s medical history of alcohol consumption (age at drinking onset, amount consumed in grams per day, and duration of the disorder) and other substances (such as tobacco and cocaine) in the 30 preceding days were recorded.

On the second day of admission, routine laboratory analysis, including blood count, liver function test, and serology for hepatitis C virus (HCV) infection, human immunodeficiency virus (HIV) infection, and hepatitis B virus (HBV) status, were analyzed. The serological tests were performed using an enzyme-linked immunosorbent assay. The HCV-positive tests were confirmed using HCV RNA (real-time polymerase chain reaction, limit of detection 50 copies/mL). The HIV-positive tests were confirmed using the Western immunoblot technique, and HBV serostatus was assessed using HBsAg, anti-HBs, and anti-HBc. To rule out autoimmune hepatitis, antinuclear antibodies were used. The patients were examined for LF according to the FIB-4 index. FIB-4 index <1.45 and >3.25 were indicative of absence of LF and advanced LF (ALF), respectively [18].
(1)$$\mathrm{FIB}\;4=\frac{\mathrm{Age}\left[\mathrm{years}\right]\times \mathrm{AST}\left[U/L\right]}{\mathrm{Platelet}\left[10^{9}/L\right]\times \sqrt{\mathrm{ALT}\left[U/L\right]}}\;*$$* AST: aspartate aminotransferase, ALT: alanine aminotransferase.

All patients underwent abdominal ultrasound at the Radiology Department, under fasting conditions, following the guidelines of the American College of Radiology. Abdominal ultrasound was used to identify liver steatosis, hepatomegaly, and signs suggestive of advanced ALD, including heterogeneous liver, portal hypertension, and cirrhosis. Portal hypertension was defined as the presence of any of the following ultrasound features: splenomegaly, enlarged portal vein, ascites, and abnormal portal vein flow. We have previously characterized the ultrasound criteria [19].

Patients who presented with clinical findings of decompensated liver disease, ultrasound findings suggestive of alcohol-related cirrhosis, diagnoses of HCV or HIV infections, and findings compatible with acute alcoholic hepatitis were excluded. Laboratory abnormalities selected to diagnose alcoholic hepatitis include bilirubin > 3 mg/dl and AST > 50 but <400 UI/L, with AST/ALT of >1.5 [20]. Figure 1 shows the steps for selecting the patients included in this study. The main analysis focused on comparing patients without evidence of LF (FIB-4 < 1.45) to those with ALF (FIB-4 > 3.25).

### 2.2. Flow Cytometry

For flow cytometry analysis, 100 μL of peripheral blood was incubated with monoclonal antibodies for 20 min at room temperature away from light exposure. Erythrocytes were extracted by lysis (BD-FACS™ lysis solution, Beckton Dickinson (BD) Biosciences, CA, USA). The samples were then washed and resuspended in flow cytometry staining buffer (BD-FACSFlow™, BD Biosciences).

Cell subpopulations were examined using combinations of the following monoclonal antibodies: anti-CD45 APC, anti-CD19 V500, anti-CD3 PerCP, anti-CD4 fluorescein isothiocyanate (FITC), anti-CD8 PE, anti-CD16 V450, and anti-CD56 V450. A total of 10,000 lymphocytes were acquired and analyzed using flow cytometry (BD-FACS Lyric, BD-Biosciences). The lymphocyte population was selected using forward- and side-scattered light, and the absolute count and relative percentage of the population of interest to total lymphocytes were determined.

The absolute counts for each cell population were calculated as (X × Y)/100, where X is the percentage of each subset and Y is the absolute lymphocyte count determined using a hematology analyzer (Beckman Coulter, FL, USA). The absolute numbers and percentages of total lymphocytes (CD45^+^), B cells (CD19^+^), T cells (CD3^+^), CD4^+^ T cells (CD3^+^CD4^+^), CD8^+^ T cells (CD3^+^CD8^+^), NKT cells (CD3^+^CD16^+^CD56^+^), NK cells (CD3^-^CD16^+^CD56^+^), and CD3^-^CD8^+^ cells were obtained. The absolute number of total lymphocytes was expressed as cells per microliter (cells/μL), and Figure 2 illustrates the strategy used for flow cytometry analysis.

Gating strategy of flow cytometric analysis of lymphocyte subpopulations in patients with alcohol use disorder (AUD). Peripheral blood mononuclear cells (PBMCs) were stained with anti-CD45 APC, anti-CD19 V500, anti-CD3 PerCP, anti-CD4 fluorescein isothiocyanate (FITC), anti-CD8 PE, anti-CD16 V450, and anti-CD56 V450. Double lymphocytes were excluded. Single lymphocytes were gated based on their scatter characteristics or forward scatter height versus forward scatter area. Total lymphocytes and T cells were then obtained based on positivity for CD45 and CD3, respectively.

### 2.3. Statistical Analysis

The results of the descriptive analysis are expressed as means ± standard deviation (SD) or median (interquartile range (IQR)) for quantitative variables and as absolute frequencies and percentages for qualitative variables. The chi-squared test was used to determine differences in categorical variables based on the LF group. To analyze the differences between the means of each group, an Analysis of Variance (ANOVA) or Kruskal–Wallis non-parametric test was used according to the normal distribution of each variable. Correlations were determined using Pearson’s or Spearman’s correlation coefficients, according to the distribution of the data. Statistical significance was set at *p* < 0.05. Statistical analyses were performed using the Stata software (version 11.1, College Station, TX, USA).

## 3. Results

### 3.1. Patients Details

The study included 136 patients (76% men) with a mean age of 49 years (IQR: 44–56), who had an AUD for 15 years (IQR: 7–20) and an alcohol consumption of 163 ± 80 g/day. Table 1 shows the sociodemographic characteristics and alcohol and substance use of the patients, according to the degree of LF. Specifically, 50 (36%) patients without LF and 25 (18%) patients with ALF exhibited similar characteristics. Hepatic steatosis was detected in 67% of all patients using ultrasound and occurred more often in patients with ALF than in patients without LF (76% vs. 48%, *p* = 0.02).

Table 2 describes the absolute values of the lymphocyte populations and laboratory parameters according to the degree of LF. Patients with ALF presented with higher AST (23 U/L vs. 90 U/L, *p* < 0.01) and ALT (23 U/L vs. 58 U/L, *p* < 0.01) levels than those without LF. However, no significant differences were observed in the levels of albumin, prothrombin, cholesterol, and triglycerides. Total bilirubin was higher in patients with ALF (0.99 mg/dL vs. 0.6 mg/dL, *p* < 0.01) than in those without LF, although the median values were within the normal limits in the two groups.

### 3.2. Count and Distribution of Lymphocyte Subpopulations

Table 2 shows that the absolute number of T, B, and NKT cells was lower in patients with ALF than it was in those without LF (*p* < 0.01). There were no significant differences in the means of absolute numbers of NK cells (170 ± 105 cells/µL vs. 208 ± 109 cells/µL) and CD3^-^CD8^+^ cells (70.1 ± 47.1 cells/µL vs. 90.3 ± 60.7 cells/µL) between the two groups.

The percentages of total lymphocytes relative to leukocytes (25.6 ± 8.1% vs. 30.9 ± 10%, *p* = 0.02) and T cells relative to total lymphocytes (77 ± 7% vs. 80.3 ± 5.6%, *p* = 0.03) were lower in patients with ALF than they were in patients without LF.

The percentage of NK cells to total lymphocytes was higher in patients with ALF than it was in patients without LF (13 ± 6.5% vs. 9.7 ± 5%; *p* = 0.02). There were no significant differences in the proportions of B, NKT, and CD3^-^CD8^+^ cells. Figure 3 shows the percentage of lymphocyte populations relative to total lymphocytes based on the degree of LF.

Lymphocyte subpopulations with respect to total lymphocytes and degree of liver fibrosis (LF). Differences in percentage of lymphocyte subpopulations in alcohol use disorder (AUD) patients without liver fibrosis (LF) (*n* = 50) and with advanced LF (ALF) (*n* = 25). Dots represent percentage of lymphocyte subpopulation for each participant.

### 3.3. Correlation between NK Cells and Lymphocyte Subpopulations

An inverse correlation was observed between the percentage of NK cells and the percentage of T cells in patients without LF (r = −0.65, *p* < 0.01) and in those with ALF (r = −0.64, *p* < 0.01). Moreover, a positive correlation was found between the percentage of NK cells and the percentage of CD3^-^CD8^+^ cells in patients without LF (r = 0.87, *p* < 0.01) and in those with ALF (r = 0.92, *p* < 0.01). No correlation was found between NK cells and the other lymphocyte populations. Figure 4 shows the correlation between NK cells and T cells, stratified in accordance with the presence of LF.

Correlations between proportion of natural killer (NK) cells, T cells, and CD3^-^CD8^+^ cells according to degree of liver fibrosis (LF). Negative correlation between percentages of NK and T cells (above in figure) in 50 patients with alcohol use disorder (AUD) without LF and 25 patients with AUD with advanced LF (ALF). Positive correlation between percentages of NK and CD3^-^CD8^+^ cells (below in figure) in 50 patients with AUD without LF and 25 patients with AUD with ALF.

## 4. Discussion

This study focused on heavy drinkers with no evidence of alcohol-related cirrhosis and revealed that the proportion of NK cells was higher in patients with ALF than it was in those without LF. The relative increase in NK cell populations correlated positively with CD3^-^CD8^+^ cells and inversely with T cells. In this study, we also observed lower NK cell counts than those previously reported in other studies [21,22]. However, it is noteworthy that the range of reference values was wide, and the role of age and sex in the count was variable. Consequently, a 4% increase for every 5 years of age and higher absolute numbers of NK cells in men compared to women have been reported [22,23]. Although the patients with ALF were older than those without LF in this study, we did not observe differences associated with age and sex (data not shown) in relation to the number and percentage of NK cells.

NK cells play an important role in the pathogenesis of liver disease. In addition to this anti-fibrotic activity, these cells promote the control of HCV and HBV infection and control of liver tumorigenesis. Through direct mechanisms of apoptosis, they are able to kill hepatic stellate cells, infected cells, and damaged cells.

For this reason, NK cells are a therapeutic target for liver fibrosis [4]. Previous studies have shown that the anti-fibrotic activity of NK cells is not only controlled by soluble factors such as interferon (IFN)-α/β, IL-12, IL-15, IL-18, and TGF-β, but also by interaction with other immune cells, such as regulatory T cells, CD4+ T cells, and macrophages [24,25,26]. Therefore, numerous treatments that increase the activation of NK cells have been proposed, including anti-TGF-β, IFN-α, and cytotherapy with macrophages type 1 [5,26,27].

We report herein a strong correlation between NK cells and CD3^-^CD8^+^ cells, which was even stronger in patients with ALF than in those without LF. Approximately 40% of NK cells express the CD8 glycoprotein, and although its role as a receptor in these populations remains to be clarified, CD8^+^NK cells have a higher cytolytic capacity than CD8^-^NK cells [28]. Furthermore, the CD8 molecule provides NK cells with a survival mechanism after lysis of target cells, potentially enabling conjugation and lysis of multiple target cells [29]. This correlation between the two populations suggests that the CD3^-^CD8^+^subset may include NK cells with a more cytotoxic profile. Although there are discrepancies regarding the cytotoxicity of NK cells in cases of chronic alcohol use, the increased cytotoxicity could favor hepatic apoptosis and the progression of ALD. Clinical studies detailing the cellular interactions that define the activity of NK cells, as well as the phenotype and function of the NK cells subset, could expand the knowledge of liver fibrosis in ALD.

The relative increase in NK cell population was related to a decrease in the proportion of T cells. CD4^+^ lymphopenia has been associated with the expression of markers of chronic activation in heavy drinkers [30] and with the expression of markers of immunosenescence in alcohol-related cirrhosis [31]. T cell lymphopenia is a common finding in patients with acute and chronic alcohol exposure [32], which may make patients susceptible to infections and may increase their morbidity [33]. The treatment of patients with AUD who have severe infections and lymphopenia is a major challenge. It has been shown that the administration of granulocyte colony-stimulating factor (G-CSF) in addition to standard treatment increases the survival of patients with either severe alcoholic hepatitis or alcoholic liver failure [34,35]. With no viable alternatives, it is necessary to explore this therapeutic strategy.

This study has limitations that are worth mentioning. First, we analyzed the global NK cells subset, and we did not analyze the intensity of the expression of CD56 and CD16 separately. The CD56^dim^ cell populations have been described as having a more cytotoxic profile and predominant in the peripheral blood. It would be interesting to analyze these subsets in future studies. In contrast, CD56^bright^ cell populations are more abundant in tissues and have longer telomerases, with presumably greater proliferative capacity than the dim populations [12]. Furthermore, functional studies analyzing the cytotoxic potential of immune cells could expand on the quantitative changes described here. Second, the results cannot generally be applied to moderate drinkers. Third, the study used a cross-sectional design, which limits the causal interpretation of the findings. Furthermore, whether the observed outcomes in this study could be reversed by abstinence from alcohol remains to be determined.

Finally, FIB-4 was more effective in the diagnosis of ALD than other serum biomarkers [36]. However, determining LF using the FIB-4 may be less accurate than using other non-invasive methods such as the Enhanced-Liver-Fibrosis-Test or FibroTest [37]. Nevertheless, FIB-4 has a greater predictive capacity in extreme stages of LF (e.g., absence of LF or ALF), correlating with the FibroTest results in patients with HCV [38]. Furthermore, FIB-4 correlates with liver-related death in patients with HCV and with hepatic complications and mortality in patients with ALD [39,40]. Moreover, it also helps to estimate the degree of LF in patients who do not have access to the healthcare system to undergo a liver biopsy or elastography [41]. A recent study demonstrated that the FIB4-index has good accuracy in predicting the risk of liver-related events (AUC of 0.821), with similar results to other biomarkers (FibroTest, Forns index, non-alcoholic fatty liver fibrosis score (NFS)). Therefore, in the absence of hepatic elastography, blood tests are a good alternative for the diagnosis and prognosis of ALD, given their accessibility in primary and secondary care [42].

Despite not performing liver biopsy or using other precise methods such as ultrasound elastography or patented biomarkers, the exclusion of patients with intermediate FIB-4 values from the analysis improved the classification of liver fibrosis. Furthermore, the availability of abdominal ultrasonography to classify patients may have minimized the misclassification of the cases in this study.

## 5. Conclusions

In the early stages of alcohol-related liver disease, alterations in the immune system are characterized by T cell lymphopenia and an inflammatory state with a greater cytotoxic response (expansion of NK and CD3^-^CD8^+^ cells), probably in response to the persistent and inadequate activation of the immune system.

## Figures and Tables

**Figure 1 jcm-11-00305-f001:**
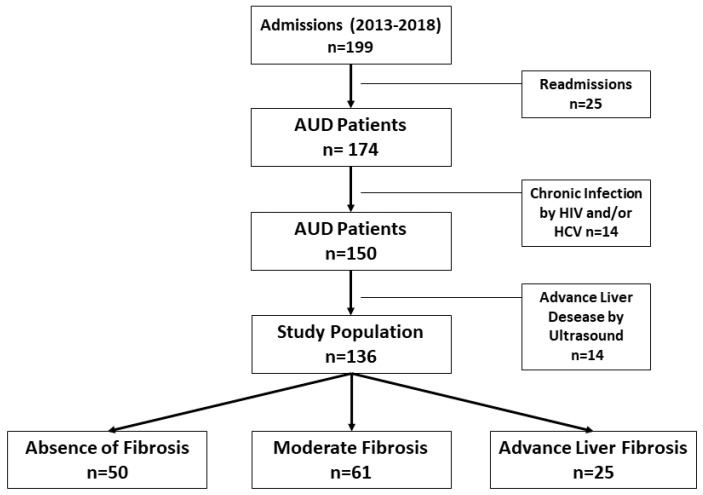
Selection of patients, exclusion criteria, and study groups according to degree of liver fibrosis (LF). AUD, alcohol use disorder; HCV, hepatitis C virus; HIV, human immunodeficiency virus.

**Figure 2 jcm-11-00305-f002:**
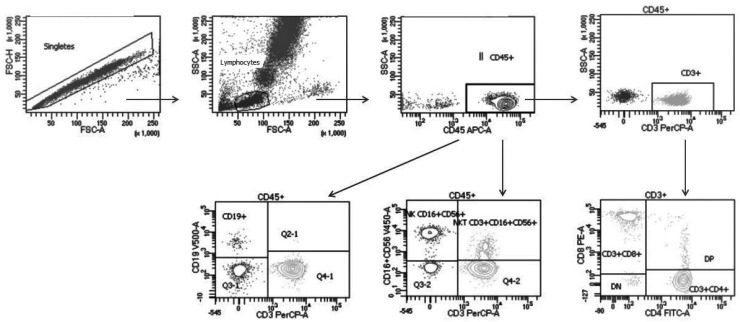
Flow cytometry strategy for analysis of lymphocyte subpopulations in study population.

**Figure 3 jcm-11-00305-f003:**
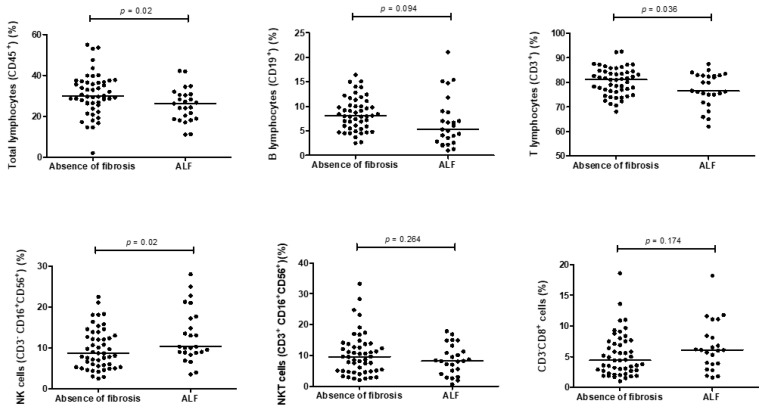
Proportion of lymphocyte subpopulations with respect to total lymphocytes and degree of liver fibrosis (LF).

**Figure 4 jcm-11-00305-f004:**
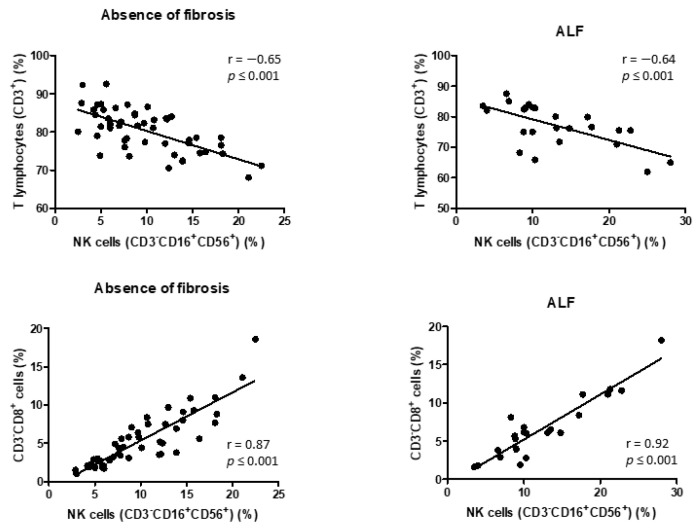
Correlations between proportion of NK cells, T cells, and CD3^-^CD8^+^ cells according to degree of liver fibrosis (LF).

**Table 1 jcm-11-00305-t001:** Sociodemographic and substance use characteristics in 136 patients with alcohol use disorders (AUD) according to the degree of liver fibrosis (LF).

Variables	Total *n* = 136 Median (RIQ)	Absence of Fibrosis *n* = 50 Median (RIQ)	Moderate Fibrosis *n* = 60 Median (RIQ)	ALF *n* = 25 Median (RIQ)	* *p*-Value
*n* (%)	104 (76)	32 (64)	83.6 (81)	21 (84)	0.073
Age (years)	49 (44–56)	47 (40–53)	51 (46–57)	52 (48–59)	<0.01
Body Mass Index (kg/m^2^, *n* = 106)	26 (23–29)	26 (22–29)	26 (23–28)	25 (22–28)	0.241
Daily alcohol consumption (g/day, *n* = 132)	140 (100–200)	138 (100–200)	160 (100–200)	140 (100–200)	0.975
Age at starting alcohol consumption (years, *n* = 96)	17 (15–20)	17 (15–20)	17 (16–20)	16 (15–18)	0.230
Duration of alcohol use disorder (years, *n* = 104)	15 (7–20)	9 (5–17)	17 (10–25)	15 (8–20)	0.03
Tobacco (%)	102 (79)	38 (79)	49 (84)	15 (65)	0.206
Cocaine use in the last 30 days (%, *n* = 133)	17 (13)	9 (18)	4 (7)	4 (16)	0.888

* Comparison of 50 patients without LF vs. 25 patients with advanced LF (ALF).

**Table 2 jcm-11-00305-t002:** Lymphocyte subpopulations and laboratory parameters in 136 patients according to the degree of liver fibrosis (LF).

	Total *n* = 136 Median (RIQ)	Absence of Fibrosis *n* = 50 Median (RIQ)	Moderate Fibrosis *n* = 60 Median (RIQ)	ALF *n* = 25 Median (RIQ)	* *p*-Value
	Lymphocyte Populations				
Total Lymphocytes (cel/µL)	1800 (1400–2300)	2050 (1700–2700)	1700 (1400–2300)	1500 (1100–1600)	<0.01
T Cells (cel/µL)	1461 (1092–1917)	1623 (1331–2252)	1457 (1014–1835)	1105 (729–1321)	<0.01
CD4^+^ T Cells (cel/µL)	925 (627–1154)	1072 (818–1350)	946 (610–1193)	602 (448–837)	<0.01
CD8^+^ T Cells (cel/µL)	416 (287–592)	448 (360–729)	424 (290–595)	329 (211–451)	<0.01
B Cells (cel/µL)	129 (84–223)	177 (124–259)	129 (88–190)	77 (39–106)	<0.01
NK Cells (cel/µL)	173 (110–276)	196 (124–264)	172 (105–282)	135 (94–196)	0.154
NKT Cells (cel/µL)	116 (74–195)	146 (76–226)	107 (74–190)	76 (37–137)	<0.01
CD3^-^CD8^+^ Cells (cel/µL)	66 (40–114)	76 (48–129)	72 (39–114)	59 (32–94)	0.161
	Laboratory Parameters				
Hemoglobin (mg/dL)	14.3 (12.9–15.5)	13.8 (12.4–15.2)	14.8 (13.4–15.4)	13.8 (13.1–15)	0.779
Platelets (×10^9^/L)	194 (153–240)	241 (207–288)	188 (157–222)	128 (105–151)	<0.01
AST (U/L)	38 (23–64)	23 (17–33)	41 (27–64)	90 (51–134)	<0.01
ALT (U/L)	28 (17–57)	23 (16–33)	32 (20–46)	58 (27–86)	<0.01
AST/ALT	1.19 (0.80–1.72)	0.88 (0.65–1.25)	1.26 (0.93–1.73)	1.9 (1.45–2.98)	<0.01
AST/ALT > 2, *n* (%)	21 (15)	1 (2)	10 (16)	10 (40)	<0.01
GGT ** (U/L, *n* = 134)	133 (37–284)	57 (31–132)	133 (52–273)	352 (77–728)	<0.01
Bilirubin (mg/dL, *n* = 135)	0.72 (0.47–1.08)	0.6 (0.43–0.9)	0.75 (0.47–1.1)	0.99 (0.82–1.2)	<0.01
Albumin (mg/dL)	39.1 (36.3–41.8)	39.1 (36–41.4)	39.1 (36.2–42)	39.4 (38.2–41.4)	0.650
Prothrombin Rate (%) (*n* = 133)	100 (92–100)	100 (95–100)	100 (89–100)	98 (87.5–100)	0.172
C-reactive Protein (mg/L) (*n* = 135)	3.1 (1–7.5)	2.6 (0.8–6.7)	4.1 (1.8–8.4)	1.7 (0.8–5.4)	0.584
Cholesterol total (mg/dL) (*n* = 135)	198 (163–239)	209 (168–247)	190 (153–224)	193 (179–239)	0.456
Triglycerides (mg/dL) (*n* = 135)	121 (83–183)	151 (99–206)	112.5 (82–175)	90 (61–121)	0.567

* Comparison of 50 patients without LF vs. 25 patients with advanced LF (ALF); ** GGT, gamma glutamyl transferase.

## Data Availability

The raw data presented in this study are available to any scientist wishing to use them for non-commercial purposes on request from the corresponding author without breaching participant confidentiality. The data are not publicly available due to privacy.

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
