# Peer review of "Increased Natural Killer Cells Are Associated with Alcohol Liver Fibrosis and with T Cell and Cytotoxic Subpopulations Change"

_jcm, 2022, doi:10.3390/jcm11020305_

Round 1
Reviewer 1 Report
The authors have correctly answered my questions.
Manuscript endorsed
Reviewer 2 Report
The authors followed the issues that I highlighted in my previous review. I don't have further comments. I accept the current form of the manuscript.
Reviewer 3 Report
All my comments were taken into account. The manuscript was revised according to my suggestions.
This manuscript is a resubmission of an earlier submission. The following is a list of the peer review reports and author responses from that submission.
Round 1
Reviewer 1 Report
In this manuscript, Zuluaga et al., aimed at evaluating the proportion of various immune cytotoxic cells in heavy drinkers without cirrhosis or with liver fibrosis. Although the study is of interest to better understand the role of such immune cells in the physiopathology of the disease, the conclusions are overinterpreted and experiments performed here leave open many questions and concerns dampen enthusiasm for the manuscript and preclude publication in “Journal of clinical medicine”
Major points :
- The authors included 136 patients in the study but it is not clear why they did not perform analyzes on patients with moderate fibrosis. It would have been interesting to analyze them also. More, a control group (no drinker) is lacking.
- As mentioned by the authors in the discussion, human natural killer (NK) cells can be subdivided into different populations based on the relative expression of the surface marker CD56. Furthermore, CD16 also helps to discriminate these subsets as the two major NK subsets are CD56bright CD16dim/− and CD56dim CD16+. The authors mentioned in their manuscript CD56+ CD16+ NK cells. Therefore, we don’t known exactly with which cells they are dealing with.
- The authors mentioned that CD8 could be express by a fraction of NK cells (discussion). However, they did not analyze its expression on NK cells in their study. The CD3- CD8+ cells they referred to is not necessary a subset of NK cells.
- The values mentioned in table 2 for NK cells and CD3- CD8+ cells did not correspond with the ones mentioned in the text.
- It would have been interesting to show CD4+ and CD8+ T cell numbers.
- Functional studies are lacking to evaluate the cytotoxic potential of the immune cells mentioned in the manuscript.
Reviewer 2 Report
The Authors give an interesting insight into the immunological background of liver fibrosis. This perspective is especially noteworthy due to its possible clinical implications and a direct connection with the treatment of liver fibrosis. I do appreciate this idea and speculations regarding found relationships between particular cell subsets. However, I miss certain discussion involving the role of NK-cells in the presentation/treatment of liver disorders. Even though, the Authors mention that increase in NK-cells and decreased population of T-cells can predispose cirrhotic patients to infections, increasing their morbidity, I would appreciate broading this context in the discussion with possible new pharmocological strategies in this field.
Reviewer 3 Report
The paper presents the problem of connections between level of natural killer cells and fibrosis in ALD and T-cell and cytotoxic subpopulations change. The paper is interesting however I have a few comments which require explanation:
- On what criteria in abdominal ultrasound signs of advanced ARLD or alcohol-related cirrhosis were diagnosed? (line 79-81). They should be listed
- There are no information, whether the patients had panendoscopy and Doppler ultrasound done. These examinations would be desirable in this study.
- Which parameters were examined to exclude the HCV, HIV, HBV infections and autoimuune hepatitis (line 87-88)
- I have some doubts whether the assessment of liver fibrosis can be based only on the FIB 4 index, without other additional imaging tests or biopsy. Authors should explain why they used only this parameter.
- The sentence „FIB-4 values…….(line 84-85) is ambigous and should be changed, beacuse FIB-4 index <1,45 suggests no liver fibrosis which is reflected later in the publication for example in the Fig. 1.
- Why the Authors did not correlate the moderate liver fibrosis with sociodemografic characteristics, level of lab tests and lymphocytes.
- Abbreviations TUA (line74) and AAH (line 88) require explanations.
- Abbreviation of alcohol-related liver disease should be ALD instead of ARLD. ALD is commonly used in literature.
